# Lipid Metabolism and Homeostasis in Patients with Neuroendocrine Neoplasms: From Risk Factor to Potential Therapeutic Target

**DOI:** 10.3390/metabo12111057

**Published:** 2022-11-02

**Authors:** Roberta Modica, Anna La Salvia, Alessia Liccardi, Giuseppe Cannavale, Roberto Minotta, Elio Benevento, Antongiulio Faggiano, Annamaria Colao

**Affiliations:** 1Department of Clinical Medicine and Surgery, Endocrinology Unit of Federico II University of Naples, 80131 Naples, Italy; 2Division of Medical Oncology 2, IRCCS Regina Elena National Cancer Institute, 00144 Rome, Italy; 3Endocrinology Unit, Department of Clinical and Molecular Medicine, Sant’Andrea Hospital, ENETS Center of Excellence, Sapienza University of Rome, 00189 Rome, Italy; 4UNESCO Chair on Health Education and Sustainable Development, Federico II University of Naples, 80131 Naples, Italy

**Keywords:** lipid metabolism, cholesterol, metabolic syndrome, neuroendocrine neoplasm, neuroendocrine tumor, cancer, cancer therapy

## Abstract

Lipid metabolism is known to be involved in tumorigenesis and disease progression in many common cancer types, including colon, lung, breast and prostate, through modifications of lipid synthesis, storage and catabolism. Furthermore, lipid alterations may arise as a consequence of cancer treatment and may have a role in treatment resistance. Neuroendocrine neoplasms (NENs) are a heterogeneous group of malignancies with increasing incidence, whose mechanisms of cancer initiation and progression are far from being fully understood. Alterations of lipid metabolism may be common across various cancer types, but data about NENs are scattered and heterogeneous. Herein, we provide an overview of the relevant literature on lipid metabolism and alterations in NENs. The available evidence both in basic and clinical research about lipid metabolism in NENs, including therapeutic effects on lipid homeostasis, are summarized. Additionally, the potential of targeting the lipid profile in NEN therapy is also discussed, and areas for further research are proposed.

## 1. Introduction

Neuroendocrine neoplasms (NENs) are a heterogeneous group of malignancies with increasing incidence, originating from cells with a neuroendocrine phenotype diffused in many different organs. The main sites of origin are the gastroenteropancreatic (GEP) tract (70%) and bronchopulmonary system (25–30%) [1,2]. NENs are mainly sporadic, but can be hereditary, associated with syndromes such as multiple endocrine neoplasia type 1 (MEN1) [3]. Diagnosis is frequently delayed because NENs are often indolent and non-functional, without specific clinical presentation or symptoms. GEP-NENs are classified according to the Ki67 index or mitotic count and differentiation in neuroendocrine tumors (NETs) with grade 1, 2 or 3 (NETs G1, G2 and G3) and neuroendocrine carcinoma (NEC). Thoracic NENs comprise typical carcinoids (TCs), atypical carcinoids (ACs), large-cell neuroendocrine carcinoma (LCNEC) and small-cell lung carcinoma (SCLC) divided according to mitotic rate and presence or absence of necrosis [4,5]. The five-year survival rate ranges from 78% to 93% in localized disease, and it decreases to 19–38% in metastatic disease [6,7]. Surgery is the best option when feasible, but a wide spectrum of systemic therapy is available for locally advanced or metastatic disease and includes somatostatin analogues (SSAs), molecular targeted therapy with everolimus and sunitinib, peptide receptor radionuclide therapy (PRRT) and chemotherapies (with different drugs such as capecitabine and temozolomide). In addition, promising data are emerging about a potential role of immunotherapy, mainly in combination with other immune check point inhibitors or with chemotherapy, in specific settings [8,9,10,11]. The goal of treatment is to improve patients’ outcomes, considering toxicities and quality of life [12,13,14,15].

It has been acknowledged that lipid metabolism is involved in tumorigenesis, as well as disease progression and treatment resistance in many common cancer types, including colon, lung, breast and prostate, through modifications of lipid synthesis, storage and catabolism [16]. Lipid alterations may represent both a promoting or accelerating factor for cancer cell growth, and their management may potentially have a role in cancer treatment [17]. In this light, different mechanisms, including overexpression of fatty acid synthase to protect cancer cells from apoptosis and lipogenesis as a support for cancer cell proliferation, have been discussed in several cancer types [17]. In addition, more recently, with the widespread use of immunotherapies, lipid metabolism has been recognized as a definite target in cancer therapy, due to its role in immunocytes reprogramming in the tumor microenvironment [18]. Nevertheless, the role of lipid alterations in NENs has never been comprehensively analyzed.

Herein, we provide an overview of the relevant literature on lipid metabolism and alterations in NENs. We performed an electronic-based search using PubMed until September 2022. The current review summarizes the available evidence both in basic and clinical research about lipid metabolism in NENs, including therapeutic effects on lipid homeostasis. Meanwhile, the potential of targeting the lipid profile in cancer therapy is also discussed, and areas for further research are proposed.

## 2. Lipids and Cancer

Lipids are a class of water-insoluble heterogeneous metabolites, with almost one million species which are involved in different cellular processes: membrane stability and processing, energy reserve, intra and extra- cellular signalling and hormone synthesis [19,20].

Lipids, in the tumor microenvironment, play not only a membrane structural function, but also a protective and communicative role. Indeed, it is known that cancer cells need a greater turnover of membrane phospholipids due to the high replication rate. The cell itself can synthesize these molecules de novo and to supply itself from the circulating pool [21,22,23]. For this reason, hypercholesterolemia has been indicated as predisposed to neoplasms. In vitro studies demonstrated that the inhibition of de novo lipid synthesis and their supply from the bloodstream is able to slow down and stop the cellular cancer replication [24,25]. This process was also reversed if the same cells were exposed to a lipid-rich environment [26]. De novo lipid synthesis by neoplastic tissues is advantageous because the integration at the level of the cell membrane takes place through an organization in clusters so as to modulate cellular signaling by favoring the interaction with tyrosine kinase receptors HER2 and IGF-1 [27,28,29,30].

Membrane lipids also play a protective role in oncological pathogenesis, as the balance among saturated, mono-unsaturated and polyunsaturated fatty acids (SFAs/MUFAs/PUFAs) gives mechanical characteristics so as to tolerate the oxidative stress of the tumor microenvironment [31]. The imbalance in favor of SFAs damages cellular plasticity and, thanks to an over expression of desaturase, the tumor cell ensures the process of desaturation and membrane plasticity. In addition, alteration in the composition of membrane SFAs/MUFAs/PUFAs on the one hand makes the cancer cell differently permeable to chemotherapeutic agents; on the other hand, it facilitates metastasis, as reported in lung cancer [32,33]. A mechanism of escape and survival has been observed, for example, within prostate cancer through the overexpression of α-methylacyl-CoA racemase, which regulates the oxidation of FAs, allowing a correct recycling to protect cellular senescence [31,34]. Furthermore, lipids play a role of energy source, since the detachment of the cancer cell from the cellular matrix imposes an energy expenditure involving the oxidation of preserved FAs. Familial hypercholesterolemia could represent a model to evaluate the relationship between lipid metabolism and cancer; however, data about NENs are still lacking. A Danish study recently analyzed the cancer risk in 221 patients with Heterozygous Familial Hypercholesterolemia (HeFH) during 25 years of follow-up. The risk of malignancies in the 117 patients carrying the LDL receptor mutation (LDL-R) was similar to the healthy control population. In contrast, HeFH patients who did not carry the mutation had a lower cancer risk than controls [35]. Another study prospectively investigated the association between familial hypercholesterolemia (FH) and lifestyle-related cancer. The FH population, compared with healthy controls, during an 8-year follow-up, appeared to have a lower risk of cigarette smoking-related cancer, although due to a lower prevalence of smoking, no difference was found for the other types of malignancies [36]. This complex role of lipids has led to studies that evaluate lipid metabolism alterations as a risk factor in cancer and lipid-lowering drugs as a protective tool. Data regarding lipid metabolism in various cancer types have been reported, including breast, colorectal and kidney. However, these data are controversial and randomized and controlled trials are needed [37,38,39,40,41]. Importantly, data about NENs are scattered and heterogeneous.

## 3. Lipid Alterations as a Risk Factor in NENs

The identification of modifiable risk factors for NENs is important not only because of their rising incidence, but also because it may be a common risk factor for the onset of non-neuroendocrine malignancies in the same patient [42,43]. The acknowledged relationship between lipid alterations and cancer, supported by in vitro studies, has led to their evaluation as a potential risk factor for cancer development. To date, only a few studies have evaluated the role of lipid alterations as risk factors in NENs.

Saturated fat intake has been proposed as a possible explanation for the positive associations between meat intake and small intestinal cancer [44]. A prospective study included 60 patients who received a diagnosis of small intestinal adenocarcinomas and 80 with small bowel NETs, with the aim to evaluate the role of meat and fat intake in relation to cancer development. This study, with an 8-year follow-up period, reported that the risk of small intestine NETs was greater in subjects in the highest compared with the lowest tertile of total fat intake (*p* trend = 0.03). Furthermore, the analyses of the continuous data confirmed a significant association between saturated fat intake and small bowel NETs. The production of bile acids from cholesterol, capable of inducing DNA damage-releasing reactive oxygen species, has been proposed as one of the potential underlying mechanisms [45].

Higher levels of cholesterol compared to healthy controls have been detected in patients with rectal NETs in a retrospective case–control study. This analysis aimed to evaluate the potentially relevant risk factors in 102 patients with rectal NETs, compared with a control group of 52.583 healthy controls at the Center for Health Promotion of the Samsung Medical Center in Korea. The control group significantly outnumbers the NET group due to the inclusion of the large number of patients undergoing a routine check-up with colonoscopy chosen as the control group. In this study, metabolic syndrome was the strongest determinant of rectal NETs. A high level of total serum cholesterol, which may also be associated with metabolic syndrome, was significantly associated with rectal NETs [46].

A cross-sectional, case–control, observational study enrolling 109 grade 1 or 2 (G1/G2) GEP-NET patients, compared with 109 healthy subjects matched by age, sex and body mass index, reported that the worsening of clinicopathological characteristics (as G2 vs. G1, progressive vs. not progressive disease, and metastatic vs. non-metastatic patients) in GEP-NET was associated with a higher presence of metabolic syndrome, non-alcoholic fatty liver disease, evaluated by the fatty liver index, and visceral adiposity dysfunction, evaluated by the visceral adiposity index. Interestingly, total and LDL cholesterol were significantly higher in GEP-NET patients than in the control group (*p* < 0.001), while HDL cholesterol was lower (*p* = 0.034) [47].

A multicenter case–control study recruited 148 GEP-NEN patients and 210 thyroid cancer patients as controls, with similar gender and age characteristics, analyzing possible risk factors for GEP-NENs, including lipid metabolism. Hypertriglyceridemia was more prevalent in GEP-NEN patients than in controls (24.3% vs. 13.8%, *p* = 0.011), even if it was not recognized as an independent risk factor. Cholesterol and HDL cholesterol levels were similar between groups. With regard to the pancreatic NEN (pNEN) subgroup, hypertriglyceridemia was significantly more prevalent than in the control group (*p* = 0.006). However, only type 2 diabetes (*p* = 0.002) and obesity (*p* = 0.007) emerged as statistically significant risk factors in the multivariate analysis [48].

Evaluation of lipid profiles was also investigated in patients with pheochromocytomas or paragangliomas (PPGLs). Indeed, hyperlipidemia in PPGLs patients may be partially related to an excess of catecholamines, and the occurrence of other malignancies has been reported [43]. The correlation between catecholamines and lipid levels was evaluated in a retrospective study enrolling 54 patients before and after surgery for PPGLs [49]. The prevalence of hyperlipidemia in these patients was 46%. On multivariate analysis, preoperative lipid levels were not significantly associated with any variable, including preoperative symptoms, body mass index, tumor location, tumor size, germline mutation or lipid-lowering medication. Preoperative serum and urinary metanephrine levels, elevated in 94.4% of patients, were associated with elevated total cholesterol and LDL levels. Interestingly, surgical resection of PPGLs was associated with an improvement in both total cholesterol, HDL and LDL levels. These findings may support a relationship between catecholamine excess and hyperlipidemia, although lipid alterations do not seem to represent a risk factor. Nevertheless, screening and monitoring of the lipid profile in PPGL patients seems appropriate [49].

The evaluation of lipid alterations as risk factors in NENs is mainly retrospective and focused on GEP-NENs in the context of metabolic syndrome. Nevertheless, the relevance of the lipid profile in NEN patients highlights the importance of evaluating the lipid profile, even when diagnosis is suspected.

## 4. Lipids Alteration in NETs

Despite the increase in knowledge of NENs’ molecular alterations, NEN onset and development are far from being understood [50,51]. Similarly, lipid alterations in NEN development are only partially explored. Neoangiogenesis may be modulated by lipid homeostasis in cancer. In particular, cholesterol metabolism and oxysterols, cholesterol-oxidized products, are known to be involved in tumor metabolism. Actually, oxysterols are capable of increasing tumor growth both directly by promoting tumor cell growth and indirectly by decreasing antitumor immune responses, inhibiting dendritic cell migration toward lymphoid organs [52].

Soncini et al. investigated whether the neutrophil-dependent angiogenic switch occurring during pNET formation was dependent on oxysterols. In this study, the authors analyzed the role of oxysterols in the spontaneous development of pNETs in RIP1-Tag2 mice, demonstrating an upregulation of the Cyp46a1 enzyme. This leads to an increased synthesis of the oxysterol 24S-HC in pancreatic islets. In addition, the oxysterol 24S-HC accumulation leads to the positioning of neutrophils close to hypoxic regions that require formation of neo-vessels, with the direct induction of neoangiogenesis. The oxysterol accumulation has possible therapeutic consequences: in fact, in this experiment, the RIP1-Tag2 mice treated with the squalene synthase inhibitor showed a significant reduction in proangiogenic neutrophils and angiogenic islets, confirming the previously described link between cholesterol metabolism and neutrophil-mediated angiogenesis [53].

Moreover, comparing the RIP1-Tag2-engineered mouse model of pNETs and their cognate human cancer, a similar transcriptomic profile has been described [54]. Cyp46a1 is overexpressed in some human pNET samples, and there is a linear correlation among Cyp46a1, VEGF and tumor diameter in G1 pNET patients. This model could represent a useful tool to test combination therapies of drugs currently used in pNET patients and cholesterol-lowering compounds endowed with a well-established pharmacologic profile (e.g., statins).

A multicenter retrospective study analyzed the serum lipid levels of patients with gastric cancer with neuroendocrine immunophenotypes (GCNEIs) in comparison with patients with adenocarcinoma or healthy controls. GCNEIs include NEC and mixed adeno-neuroendocrine carcinoma (MANEC), of which the entire or partial tumor showed neuroendocrine (NE) morphology (NEM), in addition to NE immunophenotypes, and the component of the latter should exceed 30%. The authors demonstrated that triglycerides and HDL-C levels were negatively associated with tumor size and (or) tumor progression of GC-NENM and GC- NEC, whereas a higher LDL cholesterol level could increase the risk of progressing to late tumor stages in GC-NENM patients [55].

LDL-C showed significantly different levels among GCNEIs. The LDL-C levels of GCNEIs with NEM (GC-NEC and NEC) were higher than those of the matched population with adenocarcinoma, while this has not been observed in GCNEIs without NEM (GC-NENM). Meanwhile, among GCNEIs, LDL-C was significantly higher in GCNEIs with NEM patients than in GC-NENM. These differences divided GCNEIs according to the lipid profile and were consistent with their divergent outcomes. The prognosis of GCNEIs with NEM was significantly worse than those without NEM, although without a survival difference.

On the other side, HDL cholesterol could have an indirect protective role on tumor progression, as supposed by Pereira et al. They evaluated 39 patients with well-differentiated GEP-NETs, finding that elevated IL-6 expression was associated with disease progression, and IL-6 peritumoral expression was higher in patients with low HDL cholesterol (*p* = 0.02). Even if a definite correlation between elevated IL-6 expression and low HDL cholesterol could not be demonstrated because of the presence of a cancer-related proinflammatory microenvironment, these data suggest that reducing inflammation could potentially increase HDL cholesterol levels and consequently could improve survival [56]. Importantly, low HDL levels could be a marker of inflammation.

Furthermore, NETs had higher abundances, compared to a non-cancer control group, of oxidized lysoglycerophospholipids, indicating that these tumors were under significant oxidative stress [57].

Oxidized lipids in cancer metabolism and their possible relationship with the immune response open the door to innovative therapeutic possibilities and a deeper understanding of the metabolic wiring of cancer cells [58,59,60].

## 5. Treatment-Related Lipid Alterations in NENs

Therapeutic options in NENs have considerably expanded in the last decades, and the evaluation of their safety profile should include lipid homeostasis [12]. Besides a direct effect on lipid metabolism, there could be other indirect effects due to metabolic alterations (Figure 1).

SSAs are often the first therapeutic choice in G1-G2 NETs and are widely used even in higher doses and combinations [61]. Data regarding the effect of SSAs on the lipid profile are mainly derived from studies in patients with acromegaly. In these patients, SSAs usually do not significantly modify total cholesterol and LDL cholesterol levels, whereas they may increase HDL cholesterol and decrease triglyceride concentrations [62]. In particular, 120 mg of lanreotide administered in acromegalic patients in a 48-week open-label single-arm study did not negatively affect their lipid levels [63].

Interferon alpha (IFN-a) is generally considered a second-line additive to SSA treatment in patients with refractory carcinoid syndrome or in well-differentiated NENs in clinical progression, due to its significant toxicity [64]. A retrospective study evaluated 36 patients with NENs treated with IFN-a reporting an alteration of the lipid profile in prolonged treatment, in particular, on triglyceride levels [65]. This increase in triglycerides serum levels during therapy with IFN-a has been observed in chronic viral hepatitis, in which IFN-a is more often used. A significant increase (*p* < 0.05) in total serum triglycerides was observed after the first month of therapy in the treated groups both in comparison to the basal level and to the control group [66]. Interestingly, after treatment withdrawal, triglyceride levels returned to the normal range, irrespective of treatment response, but the mechanism is still unclear [67].

Targeted therapies with everolimus and sunitinib may represent second-line therapeutic strategies in progressive NENs. Everolimus, an oral inhibitor of the mammalian target of rapamycin (mTOR), may determine hypercholesterolemia with multiple pathological mechanisms partly unknown. Indeed, inhibition of the mTOR may determine a lower lipid turnover by inhibiting the lipid neosynthesis and the sequestration of LDL from the circulation, [61,68]. In a real-world study regarding everolimus, all grades of hypercholesterolemia were reported in 13.6% [62,69], and a previous systematic review regarding the use of everolimus in extrapancreatic NETs reported hypercholesterolemia ranging from 0 to 46% of patients [12]. Therefore, a pooled analysis from the randomized, phase 3 RADIANT-3 and RADIANT-4 trials aimed to evaluate the impact on survival of hyperglycemia and hypercholesterolemia on NET patients treated with everolimus [70]. In this study, lipid alterations did not impact on progression-free survival (PFS), with similar outcomes in patients with or without hypercholesterolemia. A monocentric, retrospective study specifically evaluated plasma triglyceride and cholesterol levels with the PFS of 58 advanced pNET patients treated with everolimus. An increased risk of disease progression was reported in patients with more plasma triglycerides during the first 3 months of treatment (*p* = 0.025). In this study, the expression of two enzymes involved in fatty acid biosynthesis, acetyl-CoA carboxylase 1 (ACC1) and fatty acid synthase, were evaluated in tumor specimens by mRNA quantification and immunohistochemistry, to analyze their impact on PFS. PFS was significantly lower in patients with higher ACC1 protein expression in metastatic lesions than in patients with lower ACC1 levels (5.5 vs. 6 months; *p* = 0.039) [11].

A case of severe hypertriglyceridemia (up to 969 mg/dL) was reported in a patient with pNEN treated with a standard dose of everolimus. Impaired lipid clearance from the bloodstream and increased hepatic synthesis have been advocated as possible the underlying mechanisms involved. A rapid response to fenofibrate allowed the avoidance of everolimus reduction or discontinuation [71].

Sunitinib, a multikinase inhibitor, has been shown to induce grade 1/2 hypothyroidism in 7% of pNEN patients. As a consequence, hypothyroidism may influence the lipid profile, but no other studies specifically addressing the lipid profile in NEN patients treated with sunitinib are currently available [72].

Finally, chemotherapy has been linked to alterations in serum lipids in cancer patients [73], but the specific alterations caused by different chemotherapy regimens remain unclear [74]. Temozolomide (one of the main chemotherapy drugs used for NENs) has been demonstrated to increase fatty acid uptake both in in vitro and in vivo models of glioblastoma. The interplay between DNA alkylation and lipid uptake is not clear, but increased fatty acid uptake in glioblastoma cells may be a strategy to react to the temozolomide-induced generation of reactive oxygen species [75]. Capecitabine has also been shown to determine hypertriglyceridemia in about 4–10% of patients, but underlying mechanisms are not fully understood. Nevertheless, acute pancreatitis and a higher incidence of cardiovascular disease may occur due to capecitabine-induced hypertrygliceridemia, as an acute and chronic complication, respectively [76]. In addition, other chemotherapy regimens have been demonstrated to increase LDL cholesterol [77].

Alteration of lipid metabolism does not seem to represent a significant safety concern in NEN therapy. Nevertheless, careful monitoring and appropriate treatment are recommended, as metabolic alterations may affect disease-free survival.

## 6. Lipids as Target for Anticancer Therapies: Future Perspectives in NENs

The ability of cancer cells to expand their lipidic membrane is crucial for their survival. For this reason, some studies have focused the attention on possible molecular-targeting drugs that could interfere with lipidic membrane stability. An interesting study proposed squalene epoxidase (SQLE), a key rate-limiting enzyme involved in the cholesterol biosynthetic pathway, as a potential therapeutic target in a specific subset of aggressive and difficult to treat NENs, small-cell lung cancer (SCLC). This study demonstrated on neuroendocrine cell lines a particular sensitivity of these cells to NB-598, a known inhibitor of SQLE. In SQLE, sensitive cell lines growth defects have been observed due to the induction of apoptosis. Interestingly, the NB-598 sensitivity observed in vitro was subsequently reported in vivo in animal models, and this observation further supported the efficacy of NB-598. Importantly, the efficacy of SQLE inhibition resulted from the toxic accumulation of squalene, or squalene-derived metabolites, rather than from the inhibition of cholesterol synthesis [78].

Moreover, the potential anticancer role of other compounds named alkyl phospholipids, has been spurred forward by recent data [79]. Alkyl phospholipids are synthetic analogues of naturally occurring membrane phospholipid esters. These agents have demonstrated significant effectiveness against skin malignancies and several types of hematological malignancies. In addition, alkyl phospholipids have recently been demonstrated to be promising novel treatment methods when combined with well-established anticancer drugs [80,81].

Lipid metabolism may represent a target, even as a supportive anticancer therapy, as proposed in combination with immune checkpoint inhibitors (ICIs), currently approved drugs even for the treatment of Merkel cell carcinoma, a rare and aggressive cutaneous NEN [82]. It has been reported that the anticancer response of CD8+ T cells could be potentiated by modulating cholesterol metabolism [83]. In an animal model, the inhibition of acetyl-CoA acetyltransferase 1 (ACAT1), a key cholesterol esterification enzyme, increases the plasma membrane cholesterol level of CD8+ T cells, enhancing their proliferation and immunological response. The clustering of T cell receptors (TCRs) that bind to antigens present on the surface of cancer cells is promoted. ACAT1-deficient CD8+ T cells were better than wildtype CD8+ T cells in controlling melanoma growth and metastasis in mice [83]. These observations support a possible use of avasimibe, an oral inhibitor of ACAT1, as a boost to ICI therapy [84].

## 7. Conclusions

The mechanisms of cancer initiation and progression are complex, but the role of lipid alterations may be common across various cancer types, including NENs. Considerable efforts have been made to improve our understanding of the pathogenetic mechanism in NEN onset and progression, as well as response to therapy, including epigenetic characterization, identification of functional pathways and molecular signatures [50,51,85]. Several confounding factors may influence the lipid profile, including obesity, lipid-lowering medications and genetic predisposition, but these data endorse that NEN patients should receive a screening lipid panel periodically. New approaches in cancer therapy will probably include the modulation of lipid metabolism. Meanwhile, lifestyle measures or appropriate medications should be used to improve or reverse any modification or adverse effects on lipid metabolism.

## Figures and Tables

**Figure 1 metabolites-12-01057-f001:**
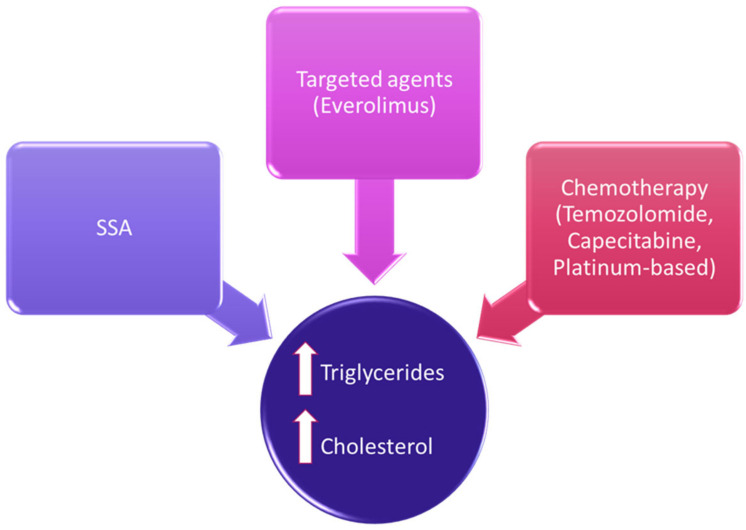
Treatment-related lipid alterations in NENs.

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
