# Peer review of "Lipid Metabolism and Homeostasis in Patients with Neuroendocrine Neoplasms: From Risk Factor to Potential Therapeutic Target"

_metabolites, 2022, doi:10.3390/metabo12111057_

Round 1

Reviewer 1 Report

The manuscript concerns a very interesting problem, which is the role of lipids in the development of neoplastic diseases and their influence on the effectiveness of the therapy. However, this manuscript needs significant improvement before it will be considered suitable for publication in a scientific journal.

Part 2 :Lipids and cancer

At the beginning of this part of the manuscript a basic information on lipids and lipoproteins is presented (lines 70 – 86). There is a so-called “book knowledge” and it should not presented in a scientific paper.

It is indicated that hypercholesterolemia is predisposing to neoplasms. However, it seems not to be well accepted. What about familiar hypercholesterolemia ? What is the relation between FH and cancer risk ? The mechanism of the association between fat and SFA intake and cancer does not seem to be fully explained by the effect on serum cholesterol levels.

Part 3: Lipid alterations as a risk factor in NENs

Lines 126-127: the text should be further elaborated as it should be 60 and 80 patients with certain diseases not 60 adenocarcinomas and 80 small bowel NETs.

Line 135 – “Higher levels of cholesterol…” – what does it mean ? higher than what ? It should be better explained especially that the group og NET patients was very small (102 patients) as compered to controls (52583 subjects).

Line 140 – “high level of cholesterol which increases the risk of metabolic syndrome” . Hypercholesterolemia is not a risk factor for metabolic syndrome as it is not the major manifestation of MS. The major risk factor is obesity, and obesity should be discussed as the risk factor as observed in MS patients disturbances in lipid metabolism are strongly related to obesity. Furthermore (lines 152-160) hypertriglyceridemia can develop in patients with obesity and DM type 2. What is known on the relation between obesity and diabetes type 2 and GEP-NENs ?

Lines 161-178 – this part must be carefully elaborated as it is hard to follow. What does it mean that is was no association between patients with PPGLs and hyperlipidemia as well as between patients with PPGLs and postoperative symptoms ,and so on.

Part 4: Lipids alterations in NETs

Editorial work is need. The text should be carefully elaborated, especially lines 216-238.  E.g. What does it mean that LDL-C showed different levels among a group of patients ? It seems there is no relation between a disease and LDL-C concentration.

Line 220-221: what described presented data: 0.515 vs -0.036 ???

Line 225 – it seems there was a correlation between LDL-C concentration and disease prognosis. It is hard to understand what “high LDL-C” mean ? What was a border level ?

Line 229- “The results ……” – what kind of results do you mean? No data are presented and no references also.

High IL-6 levels are associated with development of inflammatory process and low HDL-cholesterol can be also related to inflammation.  Therefore, low HDL-cholesterol can be only a marker. It needs discussion.

Line 234 – “NETs had higher abundance …..” higher than what ???

Line 236 – “ As additional evidence……”  What kind of additional evidence ? Where the additional evidence can be found ?

Part5: Treatment-related lipid alterations

Line 255-256- It is indicated that IFN-a treatment was reported to effect serum TG levels, but it is not clear how this effect is related to the treatment efficacy ? It is an important feature  from clinical point of view ?

261 – “by inhibiting the lipoproteins” – what does it mean ? how we can inhibit lipoproteins?  Do mean to inhibit lipoprotein synthesis?

line 294-295 – Does it mean that capecitabine significantly increases TG levels and massive hypertriglyceridemia can occur as acute effects are mentioned ?

Author Response

We thank the Editor and Reviewer for the careful and comprehensive review of our manuscript and valuable advice. In response to the reviewers’ comments and recommendations, we have revised our manuscript and answered all of the questions in a point-by-point manner. All the changes made in the revised manuscript are marked up as requested. We appreciate the reviewers' efforts and hope that this revised version is now valuable for publication.

Part 2 :Lipids and cancer

At the beginning of this part of the manuscript a basic information on lipids and lipoproteins is presented (lines 70 – 86). There is a so-called “book knowledge” and it should not presented in a scientific paper.

Thanks for this observation. We removed that part as requested.

It is indicated that hypercholesterolemia is predisposing to neoplasms. However, it seems not to be well accepted. What about familiar hypercholesterolemia ? What is the relation between FH and cancer risk ? The mechanism of the association between fat and SFA intake and cancer does not seem to be fully explained by the effect on serum cholesterol levels.

Thank you for the constructive comment. In accordance with your recommendation, we have included details about familial hypercholesterolemia (FH), which could support a better understanding of the relationship between lipids and cancer. To the best of our knowledge, unfortunately data regarding FH and NEN are still lacking. Nevertheless, we added available data about FH and other cancers’ risk (lines 113-123).

Part 3: Lipid alterations as a risk factor in NENs

Lines 126-127: the text should be further elaborated as it should be 60 and 80 patients with certain diseases not 60 adenocarcinomas and 80 small bowel NETs.

We apologize for the inaccuracy and we rephrased the text as requested (lines 138-141).

Line 135 – “Higher levels of cholesterol…” – what does it mean ? higher than what ? It should be better explained especially that the group og NET patients was very small (102 patients) as compered to controls (52583 subjects).

Thank you for the comment. In the revised manuscript, we have specified that cholesterol levels in this study were higher in NET patients compared to a control population. Furthermore, we added the reported reason of the discrepancy between the two study populations. Indeed, the authors included patients undergoing a routine check-up with colonoscopy and blood chemistry at their institute, then selected patients with a positive colonoscopy for NET from this pool, using the patients with a negative exam as control (lines 148-156).

Line 140 – “high level of cholesterol which increases the risk of metabolic syndrome” . Hypercholesterolemia is not a risk factor for metabolic syndrome as it is not the major manifestation of MS. The major risk factor is obesity, and obesity should be discussed as the risk factor as observed in MS patients disturbances in lipid metabolism are strongly related to obesity. Furthermore (lines 152-160) hypertriglyceridemia can develop in patients with obesity and DM type 2. What is known on the relation between obesity and diabetes type 2 and GEP-NENs ?

We apologize for the inaccuracy, we have rephrased the sentence specifying that high level of cholesterol may be associated with metabolic syndrome (lines 157-159). However, the relation between GEP NEN, diabetes type 2 and obesity goes beyond the aim of this study, as it would require a comprehensive and detailed analysis. We thank for this suggestion which could be the subject of further studies.

Lines 161-178 – this part must be carefully elaborated as it is hard to follow. What does it mean that is was no association between patients with PPGLs and hyperlipidemia as well as between patients with PPGLs and postoperative symptoms ,and so on.

We apologize for the lack of clarity. We have now revised the entire description of the study (lines 191-200), highlighting the importance of monitoring lipid profile even in rare NET as PPGL. Indeed, a positive correlation between preoperative elevated metanephrine and hyperlipidemia has been reported, together with an improvement of hyperlipidemia after surgery for PPGL.

 Part 4: Lipids alterations in NETs

Editorial work is need. The text should be carefully elaborated, especially lines 216-238.  E.g. What does it mean that LDL-C showed different levels among a group of patients? It seems there is no relation between a disease and LDL-C concentration.

We apologize. We have carefully revised this section with the aim to be more precise and accurate (lines 245-258).

Line 220-221: what described presented data: 0.515 vs -0.036 ???

Thanks for this question. In this revision we removed this confounding data.

Line 225 – it seems there was a correlation between LDL-C concentration and disease prognosis. It is hard to understand what “high LDL-C” mean? What was a border level?

We apologize for the difficult reading. In the absence of a threshold, we rephrased the sentence trying to clarify the results (lines 253-257).

Line 229- “The results ……” – what kind of results do you mean? No data are presented and no references also.

We apologize for the inaccuracy. We have now specified in the text the data presented and the reference (lines 260-262).

High IL-6 levels are associated with development of inflammatory process and low HDL-cholesterol can be also related to inflammation.  Therefore, low HDL-cholesterol can be only a marker. It needs discussion.

Thanks for the constructive comment. In accordance with your recommendation, we have included the proposed explanation of the relationship between HDL cholesterol and inflammation (lines 265-269), although we agree that low HDL cholesterol could only be a marker.

Line 234 – “NETs had higher abundance …..” higher than what ???

We apologize for the inaccuracy. We have now specified in the text that higher abundance was compared to healthy control population (line 270).

Line 236 – “ As additional evidence……”  What kind of additional evidence ? Where the additional evidence can be found ?

Thanks for this observation. We revised the sentence according to your suggestion (lines 273-276).

Part5: Treatment-related lipid alterations

Line 255-256- It is indicated that IFN-a treatment was reported to effect serum TG levels, but it is not clear how this effect is related to the treatment efficacy ? It is an important feature  from clinical point of view ?

Thanks for this observation. We added some evidence on relationship between IFN-a treatment and serum TG levels (lines 294-300; Ruiz-Moreno, Increase in triglycerides during alpha-interferon treatment of chronic viral hepatitis, 1992; Tominaga association between capacity of interferon-alpha production and metabolic parameter, 2010) in chronic viral hepatitis, since IFN-a is now rarely used in NENs, as somatostatin analogs are preferred. Nevertheless, the relationship with treatment efficacy is not clear.

261 – “by inhibiting the lipoproteins” – what does it mean? how we can inhibit lipoproteins?  Do mean to inhibit lipoprotein synthesis?

Thanks for this observation. We revised the sentence (lines 305-307) and we added a reference to clarify this inhibition of lipid neosynthesis (Laplante, An emerging Role of mTOR in Lipid Biosynthesis, 2009).

line 294-295 – Does it mean that capecitabine significantly increases TG levels and massive hypertriglyceridemia can occur as acute effects are mentioned?
Thanks for this observations. Hypertriglyceridemia is a rare adverse events in patients treated with capecitabine, whose mechanism are not fully understood. Following your request, we have specified in the text that acute complications may be related to capecitabine induced hypertriglyceridemia and we have detailed those complications (lines 342-347).

Reviewer 2 Report

General comments to the paper entitled

Lipid metabolism and homeostasis in patient with neuroendocrine neoplasms: from risk factor to potential therapeutic target

The author’s aim was to provide an overview of the literature on the relationship between lipid metabolism and cancer. The paper presents basic and clinical research relating to lipid metabolism in neuroendocrine neoplasms (NENs).

In “Lipids and cancer” section they conclude the data regarding the lipid metabolism in various cancer types are controversial and heterogeneous, which conclusion is supported by the cited papers.

In “Lipid alterations as a risk factor in NENs” section the authors conclude only a few studies are available that are related to lipid alteration as a risk factor in NENs. This part of the paper presents data for the correlation between lipid alteration and cancer but the correlation was even stronger with metabolic syndrome, type 2 diabetes, and obesity.

In “Lipids alteration in NETs” section based on the cited papers the authors cannot make any conclusive conclusion which is not the fault of the authors but the limited available data.

In “Treatment-related lipid alterations in NENs” authors cite relevant papers proving the effect of applied cancer drugs on lipid composition.

A very important paper is cited which proves that Temozolomide increases fatty acid uptake both in vitro and in vivo.

How can the authors explain the efficacy of Temozolomide when the limitation of membrane synthesis could be a strategy to limit tumor growth, but the cited paper suggests Temozolomide supports the fatty acid intake of cancer cells?

line 287-289 is repeated in line 298-300

The authors missed interpreting the research data relating to the ketogenic diet. The increasing number of papers prove the efficacy of the ketogenic diet in cancer therapy by increasing the daily fatty acid intake. This approach focuses on the role of metabolism in cancer development. I think talking about lipids metabolism relating to cancer growth is important to present this side of research which may provide more convincing and conclusive conclusions.

Author Response

We thank the Editor and Reviewer for the careful and comprehensive review of our manuscript and valuable advice. In response to the reviewers’ comments and recommendations, we have revised our manuscript and answered all of the questions in a point-by-point manner. All the changes made in the revised manuscript are marked up as requested. We appreciate the reviewers' efforts and hope that this revised version is now valuable for publication.

Reviewer 2

In “Lipids and cancer” section they conclude the data regarding the lipid metabolism in various cancer types are controversial and heterogeneous, which conclusion is supported by the cited papers.

Thanks for this comment.

In “Lipid alterations as a risk factor in NENs” section the authors conclude only a few studies are available that are related to lipid alteration as a risk factor in NENs. This part of the paper presents data for the correlation between lipid alteration and cancer but the correlation was even stronger with metabolic syndrome, type 2 diabetes, and obesity.

Thanks for this comment.

In “Lipids alteration in NETs” section based on the cited papers the authors cannot make any conclusive conclusion which is not the fault of the authors but the limited available data.

Thanks for your understanding.

In “Treatment-related lipid alterations in NENs” authors cite relevant papers proving the effect of applied cancer drugs on lipid composition.

Thanks for this comment.

A very important paper is cited which proves that Temozolomide increases fatty acid uptake both in vitro and in vivo.

How can the authors explain the efficacy of Temozolomide when the limitation of membrane synthesis could be a strategy to limit tumor growth, but the cited paper suggests Temozolomide supports the fatty acid intake of cancer cells?

Thanks for this interesting observation, In the cited paper the author admit that the interplay between DNA alkylation and lipid uptake is not clear, but increased fatty acid uptake in glioblastoma cells may be a strategy to react to temozolomide-induced generation of reactive oxygen species. We have added this information in the revised text (lines 340-342).

line 287-289 is repeated in line 298-300

We apologize for the mistake and we have removed the repeated lines.

The authors missed interpreting the research data relating to the ketogenic diet. The increasing number of papers prove the efficacy of the ketogenic diet in cancer therapy by increasing the daily fatty acid intake. This approach focuses on the role of metabolism in cancer development. I think talking about lipids metabolism relating to cancer growth is important to present this side of research which may provide more convincing and conclusive conclusions.

Thanks for this insightful comment. Ketogenic diet has been reported to be a promising approach in several types of cancer types, nevertheless real world data in NENs are still lacking. For this reason, we cannot actually include any detailed information. Nevertheless, we thank for this suggestion which could be the subject of further studies specifically focusing on this topic.

Reviewer 3 Report

It is a well-organized and well-written review.  Some typos to correct: 

Page 3 Line 138: a control group of 52.583 healthy controls;

Page 3 Line 151:  was lower (P 0.034)

Page 4 Line 159:  only type 2 diabetes (p 0.002) and obesity (p 0.007)

Page 6 Line 284: Revise these sentences -  Sunitinib, a multikinase inhibitor has been shown to induce grade 1/2 hypothyroidism in 284 the 7% of pNENs patients, consequentially it may influence lipid profile no other specific 285 studies on lipid profile are currently available [69].

Author Response

We thank the Editor and Reviewer for the careful and comprehensive review of our manuscript and valuable advice. In response to the reviewers’ comments and recommendations, we have revised our manuscript and answered all of the questions in a point-by-point manner. All the changes made in the revised manuscript are marked up as requested. We appreciate the reviewers' efforts and hope that this revised version is now valuable for publication.

It is a well-organized and well-written review.  Some typos to correct:

Thanks for your comment.

Page 3 Line 138: a control group of 52.583 healthy controls;

As requested by reviewer 1, we have specified that the control group significantly outnumbers the NET group due to the inclusion of the large number of patients undergoing a routine check-up with colonoscopy chosen as control group.

Page 3 Line 151:  was lower (P 0.034)

We apologize for the inaccuracy. We have now corrected (line 168).

Page 4 Line 159:  only type 2 diabetes (p 0.002) and obesity (p 0.007)

We apologize for the inaccuracy. We have now corrected (line 176).

Page 6 Line 284: Revise these sentences -  Sunitinib, a multikinase inhibitor has been shown to induce grade 1/2 hypothyroidism in 284 the 7% of pNENs patients, consequentially it may influence lipid profile no other specific 285 studies on lipid profile are currently available [69].

Thanks for pointing this out. We have revised the sentence (lines 329-332) explaining that lipid profile may be altered by the sunitinib-induced hypothyroidism.

Round 2

Reviewer 1 Report

The manuscript was found significantly improved. No additional comments

Reviewer 2 Report

I accepted the author's response.